# Health and economic burden of foodborne zoonotic diseases in Amhara region, Ethiopia

**Sefinew Alemu Mekonnen**[1]*, **Agegnehu Gezehagn**[2], **Adugna Berju**[1], **Belete Haile**[1], **Haileyesus Dejene**[1], **Seleshe Nigatu**[1], **Wassie Molla**[1], **Wudu Temesgen Jemberu**[1]

**1** Department of Veterinary Epidemiology and Public Health, College of Veterinary Medicine and Animal Sciences, University of Gondar, Gondar, Ethiopia, **2** Central Gondar Zone Animal Quarantine Center, Gondar, Ethiopia

* asefinew@gmail.com

## Abstract

Diseases from food of animal origin are common health problems in Ethiopia. A cross-sectional study was carried out to estimate health and economic burden, and to identify demographic factors associated with community awareness of foodborne zoonotic diseases in Amhara region, Ethiopia. Data was collected from 435 households in three towns: Gondar, Lalibela and Debark. A retrospective data was also collected from health records in each town. The health burden due to zoonotic diseases was estimated at 0.2, 0.1 and 1.3 DALYs per household per year and at 73.2, 146.6 and 1,689.5 DALYs out of 100,000 populations per year in Gondar, Lalibela and Debark, respectively. The overall health burden due to foodborne zoonotic diseases (aggregated over the 435 households in the three towns) was estimated to be 89.9 DALYs per 100,000 populations per year. The economic impact of foodborne zoonotic diseases in the three towns of Amhara regional state was 278.98 Ethiopian Birr (ETB) (1ETB = 0.025 US Dollar) per household per year and 121,355.68 ETB per year. Costs of preventive measures followed by costs of patients' time made the highest contribution while costs of diagnosis made the lowest contribution to the total economic burden of foodborne zoonotic diseases. From a total of 435 respondents, 305 (70.1%) had known the presence of zoonotic diseases. Level of education, number of families in the house and income were highly associated with awareness of zoonosis. Although majority of respondents had known zoonotic diseases exists (70.1%) and disease can be acquired from animal source food (63.2%), the health and economic burden associated to foodborne zoonotic diseases are still high. Therefore, changing mindset and practical training aiming in controlling foodborne zoonotic diseases may be suggested to the community in the health improvement extension service.

## 1. Introduction

Zoonotic diseases are diseases transmissible between people and animals. Nearly two-thirds of human infectious diseases exerting heavy public health and economic burden to the global community originate from animals [1, 2]. In the absence of proper care of the relation between

**Funding:** This study was financially supported by University of Gondar, Ethiopia with grant Ref. number 383/10/2019. The funders had no role in study design, data collection and analysis, decision to publish, or preparation of the manuscript.

**Competing interests:** The authors have declared that no competing interests exist.

animal and human populations, zoonotic diseases can lead to a serious risk of public health with huge economic consequences [3]. Zoonotic diseases threaten the health and productivity of animals, the livelihood of people, and cause illness and death in consumers [4]. Worldwide, the direct cost of zoonotic diseases over the last decade has been estimated to be more than $20 billion with over $200 billion indirect losses [5]. Zoonotic diseases and diseases recently emerged from animals have been estimated to contribute more than a quarter of the disability-adjusted life years (DALYs) lost to infectious diseases in low income countries such as sub-Saharan Africa [6]. In these countries, however, the economic and health burden associated to zoonotic disease is often given low attention and is largely unknown [4]. In order to obtain a better understanding of their health impact, calculating summary measures of population health such as the DALY is necessary. The attention given to zoonotic diseases has however focused more on emerging zoonotic diseases that pose global economic and health threats than endemic zoonotic diseases [7–9]. Quantitative data on health and economic burden of zoonotic diseases may contribute in changing mindset to make informed decision on prevention and control of endemic zoonotic diseases.

Foodborne diseases have enormous impacts on the health and livelihoods of people around the globe and are of great concern to consumers, producers, and policymakers. Although many articles have been published on (foodborne) zoonotic diseases worldwide [10–14]), specific information on health and economic impact of (foodborne) zoonotic diseases in developing countries is limited [4]. So, in order to get a better understanding of the impacts of zoonotic diseases specifically for the situation in developing countries, estimates of the health and economic burden of foodborne zoonotic diseases are needed. By estimating the burden of foodborne zoonotic diseases, awareness of the economic and health consequences of these diseases can be created, which is a first step for prevention and an important part of motivating policy makers to take preventive measures against foodborne zoonotic diseases. Particularly, the estimation of the burden of foodborne zoonotic diseases in households can be helpful, as it provides households with estimate to assess their own situation.

Ethiopia has the second largest human population in Africa and the largest in livestock population in the continent [15]. About 80% of Ethiopians are dependent on agriculture and have direct contact with livestock or other domestic animals which creates opportunity for infection and spread of zoonotic diseases between humans and animals [4]. This and having a large population of low income livestock farmers, Ethiopia also ranks very high in the health burden of zoonotic diseases [13]. The situation is very much pronounced in rural communities of Ethiopia where animals and humans share the same microenvironment including a shelter [16]. Besides, the handling and widespread habit of raw beef consumption in Ethiopia can favor the spread of foodborne zoonotic infection. Raw meat without appropriate temperature control is available in open-air in local retail shops for sale and minced meat, 'Kitfo' in Ethiopian Amharic language, is served as raw or under cooked at restaurants.

There are a lot of foodborne diseases in Ethiopia [4]. Salmonellosis is one of the most common foodborne bacterial zoonotic diseases identified from food animals in Ethiopia [17]. Non-typhoidal *Salmonella* represents an important human and animal pathogen worldwide [18]. In humans, in addition to concern about foodborne zoonotic diseases caused by *Salmonella* organisms, concern has also been raised about the impact of acquired antimicrobial resistance transferred among these organisms [19]. Listeriosis is an important emerging foodborne bacterial zoonotic infection worldwide. Ready-to-eat food-mediated listeriosis infection in humans has been documented by several workers from different parts of the world [20, 21]. In Ethiopia, Molla et al. [22] demonstrated widespread occurrence and distribution of *L. monocytogenes* and other *Listeria* species in retail meat and milk products in Addis Ababa. Despite the potential threat to human health posed by listeriosis, there is scarcity of data on the public

health and economic burden associated to this disease. From, helminthes, taeniasis caused by *Tinea saginata* is common infection associated to eating raw or undercooked beef in many sections of human population in the country [23].

Relatively "many articles" are published on the epidemiology of foodborne zoonotic diseases in Ethiopia [4, 16, 24–31]. However, to our knowledge, there is no literature on the health and economic burden of foodborne zoonotic diseases [4]. The present study aimed at estimating the impact of foodborne zoonotic diseases on health and economy, and to identify factors associated with awareness of the people about these diseases in Amhara region, Ethiopia.

## 2. Materials and methods

### 2.1. Study area

The study was conducted in three towns selected from Amhara regional state: Gondar, Debark and Lalibela. These towns were selected because they are destinations for large number of tourists. The 2014–2017 projected populations, projection made based on the results of National population and Housing Census of Ethiopia conducted in May 2007, for the three towns were 306, 246 for Gondar, 30,781 for Debark and 27, 200 for Lalibela [32].

### 2.2. Study population and animal source food consumption habits

As there was no previous study that estimated health and economic burden of foodborne zoonotic diseases in Ethiopia, the sample size was not estimated based on previous study. The sample size was 435 households by our expertise estimate of representativeness and taking time and logistical considerations into account. This sample size was allocated proportional to the size of the households for each of the towns: 286 to Gondar, 71 to Lalibela, and 78 to Debark. Households were randomly selected and heads of households were approached; 435 heads of households who volunteered to participate in the study were included.

The studied households consume both cereals and animal products. Overall diets are dominated by grains and that both per capita consumption and food budget shares of animal source food are low. Nowadays, the quantity of animal source food consumption is increasing although the increase is modest because of relatively higher price of animal products. Consumption of animal source food is also seasonal. Orthodox Christians don't consume animal source food during fasting season and on Wednesdays and Fridays except the 50 days running from Easter. Animal source food consumption peak is associated with major religious events [33]. No such seasonality is seen during the Muslims' major fasting season of Ramadan. Both religious groups are dependent on limited types of animals for meats due to cultural taboo for some food animals such as pigs. The per capita consumption in towns is higher than the per capita consumption in rural areas linked to income differences and associated resulting changes in household preferences [34]. A cow or an ox is commonly butchered for the sole purpose of selling and most people buy raw meat from butchers and cook and consume within their home. In special occasions, people have a cultural ceremony of slaughtering cow or ox and sharing among the group. In addition to consuming completely or partly cooked meat, eating raw meat is a common practice. At the larger butcheries, many frequently sit and eat traditional meal of higher quality raw meat [35].

### 2.3. Design of the questionnaire and data collection

A questionnaire was prepared to collect data on demography, household health expenses, treatment service, time spent on seeking health service, and risks associated to diarrhea disease. Most of the questions were closed questions to make ease of the analysis. The original

questionnaire was prepared in English and translated into local language (Amharic), and subsequently translated back into English by an external translator to validate the translation. The questionnaire was pilot-tested by administering to 5 people that have similar characteristics to the study participants outside the study. The questionnaire was corrected based on the response of the pilot test. The English version of the questionnaire is provided in S1 Questionnaire.

The questionnaire was administered by face-to-face interviews to the 435 respondents recruited from the study towns. In addition, a retrospective data was collected from records of health centers serving the study towns. In the retrospective data collection, cases with signs of diarrhea referred for laboratory confirmation in the last two years were sought from patient records of two health centers in each of the three towns. In accordance with the local legislation and institutional requirements, ethical review and approval was not required for such questionnaire study. A verbal informed consent was obtained from respondents before the start of the interview and the data was analyzed and reported anonymously.

## 2.4. Data management and analysis

The health and economic impact of foodborne zoonotic diseases was estimated based on the number of cases of diarrhea obtained from respondents and the proportion of diarrhea case caused by foodborne zoonotic diseases obtained from health center records. The proportion of diarrhea cases caused by foodborne zoonotic diseases (0.49) derived by dividing the number of diarrhea caused due to confirmed foodborne zoonotic diseases (653) to the total number of diarrheic cases confirmed by laboratory (1332) in the last one year. As respondents were not willing to tell their household size, average household size of 4.8 for the region was used from secondary sources [36]. Data used to estimate the impact of foodborne zoonotic diseases on health and economy and to identify factors associated with awareness of the people about these diseases in Amhara region, Ethiopia are available in S1 Table.

## 2.5. Calculation of impacts of foodborne zoonotic diseases on health

Health impact of foodborne zoonotic diseases was estimated by summary disease measure of disability adjusted life years (DALY) which is sum of years of life lost (YLL) due to premature mortality and years lived with disability (YLD) [37, 38].

**Years of life lost.** Years of life lost due to death associated to foodborne zoonotic diseases was estimated from expected years of life lost at death derived from the standard life table for Ethiopia as given by WHO and from the number of deaths due to foodborne zoonotic diseases as

$$YLL_i = \sum_{i=Birth}^{n}(N_i^D * E_i) \tag{1}$$

Where

$YLL_i$ = Years of life lost due to death associated with foodborne zoonotic disease in household with age $i$

$N_i^D$ = Number of human death in household with age $i$

$E_i$ = Life expectancy of the concerned category of age i

**Years lived with disability.** Years lived with disability was estimated separately for disability due to the diseases and due to the medication.

**Years lived with disability due to diarrhea.** Years lived with disability due to foodborne zoonotic disease associated diarrhea was estimated from number of diarrheic patients, average duration of stay with the diarrhea and disability weight of 0.17 for diarrhea, averaged from the

disability report for mild, moderate and sever cases of diarrhea from Salomon et al. [39] as

$$YLD_D = N^{PT} * T * DW_D \qquad (2)$$

Where:

$YLD_D$ = Years lived with disability due to foodborne zoonotic disease associated diarrhea in household

$N^{PT}$ = Number of diarrheic patients in household

$T$ = Average duration of stay with diarrhea associated to the foodborne zoonotic diseases in household

$DW_D$ = Disability weight of diarrhea

**Years lived with disability due to medication.** This was estimated from the number of patients treated for foodborne zoonotic diseases associated diarrhea, average duration of treatment and disability weight of medication. Disability weight of 0.07 for daily medication for generic uncomplicated diseases estimated by Haagsma et al. [40] was used.

$$YLD_{Mi} = N_i^{pT} * T_i * DW_M \qquad (3)$$

Where:

$YLD_{Mi}$ = Years lived with disability due to medication of diarrhea associated with foodborne zoonotic disease in household $i$

$N_i^{PT}$ = Number of patients treated for diarrhea associated with foodborne zoonotic diseases in household $i$

$T_i$ = Average duration of treatment of with diarrhea associated to the foodborne zoonotic diseases in household $i$

$DW_M$ = Disability weight of medication from the literature for generic uncomplicated diseases

Total DALYs in household was the summation of years of life lost due to death, years lived with disability due to the diseases and years lived with disability due to medication of diarrhea associated to foodborne zoonotic diseases as

$$\text{DALY} = \sum_{i=Birth}^{n}(YLL_i) * YLD_D + YLD_M \qquad (4)$$

Where:

$YLL_i$ = Years of life lost due to death associated to foodborne zoonotic disease of individual with age $i$

$YLD_D$ = Years lived with disability due to foodborne zoonotic disease associated diarrhea

$YLD_M$ = Years lived with disability due to medication associated to foodborne zoonotic disease associated diarrhea

## 2.6. Estimating economic impacts of foodborne zoonotic diseases

The economic impact of foodborne zoonotic diseases for study participant households was aggregated from the cost of diagnosis, cost of drug, cost of informal care takers and cost of patients' time per year. Economic impact due to illness falls into two broad categories, namely direct and indirect costs. Direct costs are costs incurred for health care services such as costs of diagnosis and costs for drugs. Indirect costs are opportunity costs due to productive working time losses resulting from illness and to other healthy members of household [41].

**Costs of diagnosis of foodborne zoonotic diseases.**

$$C_i^{Di} = N_i^{PT} * P_i^{Di} \tag{5}$$

Where:

$C_i^{Di}$ = Costs of diagnosis of foodborne zoonotic diseases in household $i$

$N_i^{PT}$ = Number of patients treated for foodborne zoonotic disease in household $i$

$P_i^{Di}$ = Price of diagnosis per case in household $i$

**Costs of drugs.**

$$C_i^{Dr} = N_i^{PT} * P_i^{Dr} \tag{6}$$

Where:

$C_i^{Dr}$ = Costs of drug to treat a case associated to foodborne zoonotic disease in household $i$

$N_i^{PT}$ = Number of patients treated for foodborne zoonotic disease in household $i$

$P_i^{Dr}$ = Price of drug per case in household $i$

**Costs of informal care takers.**

$$C_i^{IC} = N_i^{PT} * T_i^{IC} * P_i^{Lh} \tag{7}$$

Where:

$C_i^{IC}$ = Costs of informal care takers in household $i$

$N_i^{PT}$ = Number of patients treated for foodborne zoonotic disease in household $i$

$T_i^{IC}$ = Time taken by informal care takers to nurse a patient in household $i$

$P_i^{Lh}$ = Price of labor per hour (25 Ethiopian Birr) in Amhara region. The official exchange rate during the study year was One ETB for 0.025 USD.

**Costs of patients' time.**

$$C_i^{TP} = ((N_i^{PT} * T_i^{IT}) + (N_i^{PuT} * T_i^{IPuT})) * P_i^{Lh} \tag{8}$$

Where:

$C_i^{PT}$ = Costs of patients' time associated to illness in household $i$

$N_i^{PT}$ = Number of patients treated for foodborne zoonotic disease in household $i$

$T_i^{IL}$ = Productive working time losses resulting from illness and treatment in patients treated in household $i$

$N_i^{PuT}$ = Number of patients untreated for foodborne zoonotic disease in household $i$

$T_i^{IPuT}$ = Productive working time losses resulting from illness in patients untreated in household $i$

$P_i^{Lh}$ = Price of labor per hour (25 Ethiopian Birr) in Amhara region

## 2.7. Analyzing factors associated with awareness of foodborne zoonotic diseases

Descriptive statistics such as frequency distribution and percentages were used to summarize the data. Factors associated with awareness of foodborne zoonotic diseases were analyzed using logistic regression. The dependent binary variable was awareness of foodborne zoonotic disease (yes or no). Independent variables were study town (Gondar, Lalibela and Debark), gender (male and female), household size (one person or >one person), age in years (<35, >35 to 50 and >50), level of education (illiterate, general education, college education), and income per month (<2500 Ethiopian Birr (ETB), 2501–5000 ETB, 5001–7500 ETB and >7500 ETB.

Independent variables were screened using univariable logistic regression. Pair-wise correlations between independent variables were evaluated using the Spearman Rank correlations for variables significant in the univariable analysis. If two variables had a correlation coefficient of ≥0.7, only one of the variables was included in the multivariable analysis. Variables statistically significant at $P < 0.05$ in the univariable analyses were tested starting from the full model by removing one variable at a time in the same multivariable logistic regression models using backward reduction. Variables in multivariable models with $P < 0.05$ from the Wald test were retained. All two-way interactions between variables in the final multivariable models were tested. Confounding was checked during the model building process by evaluating the change in the coefficients of other variables when a variable was eliminated from the models. If this change in beta estimate was >30%, the variable was considered a confounder. The analyses were done using Stata statistical software release 14 (Stata Corp LLC, USA).

## 3. Results

Majority of respondents, 298 (68.5%) were males. The youngest and the oldest age of the respondents were 11 and 66 years, respectively while their average age was 40 years. Majority of respondents, 355 (81.6%) attended college education while 26 of them were illiterate, and 54 attended general education. Majority of households 265 (61%) had more than one person in the household while 170 of households had one person in the household.

### 3.1. Health and economic burden of foodborne zoonotic diseases

The health and economic impact estimates were based on three diarrhea causing foodborne zoonotic diseases: salmonellosis, listeriosis and taeniasis which were confirmed in health centers of the three towns: Gondar, Lalibela and Debark in North West Ethiopia. The minimum, average and maximum values of the most important input items for the health and economic impact estimate are summarized in Table 1.

An average annual human mortality rate of 0.01 was recorded associated to foodborne zoonotic diseases. The minimum, average and maximum income of respondents were 500 ETB, 4,652.6 ETB and 12,000 ETB, respectively. Eighteen percent of households did not visit health

**Table 1. Summary of the minimum, average and maximum annual values of parameters used in estimating health and economic burden of foodborne zoonotic diseases (N = 435).**

| Model parameters | Minimum | Average | Maximum |
|---|---|---|---|
| Number of diarrheic patients per household | 1 | 1.5 | 3 |
| Number of patients treated per household | 0 | 0.9 | 2 |
| Costs of diagnosis (ETB[1]) | 5 | 36.4 | 350 |
| Costs of treatment and drug (ETB) | 20 | 82 | 500 |
| Duration of treatment (days) | 0 | 1.7 | 9 |
| Sick leave time (days) | 0 | 4.2 | 33 |
| Waiting for diagnosis (hours) | 0 | 3.2 | 12 |
| Time taken of informal care takers (days) | 0 | 0.7 | 6 |
| Number of dead people per household | 0 | 0.03 | 1 |
| Age of death of individuals (years) | 0 | 0.08 | 5 |
| Days elapsed before seeking treatment (Days) | 0 | 2.24 | 7 |
| Costs of prevention of diarrhea (ETB) | 0 | 431 | 3,000 |

[1]Ethiopian Birr.

**Table 2. Annual health burden due to foodborne zoonotic diseases in disability adjusted life years per 100,000 population in Gondar, Lalibela and Debark towns in Amhara region, Ethiopia.**

| Source of health burden | Gondar | Lalibela | Debark | Overall |
|---|---|---|---|---|
| Years of life lost | 38.9 | 0 | 1,530.2 | 66 |
| Disability due to medication | 6.5 | 37.5 | 23.7 | 4.6 |
| Disability due to foodborne associated diarrhea | 27.8 | 109.1 | 135.6 | 19.3 |
| Total DALY[1] | 73.2 | 146.6 | 1,689.5 | 89.9 |

[1]Disability adjusted life years.

centers to seek treatment for diarrhea. Costs of diagnosis, costs of treatment and drug, and costs of prevention of diarrhea varied largely between households.

### 3.2. Public health burden of foodborne zoonotic diseases

Based on the number of respondents in each of the towns, the health burden due to foodborne zoonotic diseases in Gondar, Lalibela and Debark towns were 0.2, 0.1 and 1.3 DALYs per household per year, respectively. Further, the health burden was estimated at 73.2, 146.6 and 1,689.5 DALYs out of 100,000 populations per year in Gondar, Lalibela and Debark towns, respectively. The overall health burden due to foodborne zoonotic diseases (aggregated over the 435 households in the three towns) was estimated to be 89.9 DALYs per 100,000 populations per year (Table 2).

### 3.3. Economic impact of foodborne zoonotic diseases

The economic impact of foodborne zoonotic diseases in the three towns of Amhara regional state, in average, was 278.98 ETB per household per year. The overall economic burden due to foodborne zoonotic diseases (aggregated over the 435 households in the three towns) was estimated to be 121,355.68 ETB per year. Of the different cost factors, costs of preventive measures' made the highest contribution while costs of diagnosis made the lowest contribution to the total economic impact of foodborne zoonotic diseases (Table 3).

### 3.4. Awareness of foodborne zoonotic diseases

From a total of 435 respondents, 305 (70.1%) knew the presence of zoonotic diseases and 275 (90%) of them knew zoonotic diseases can be acquired from food. The awareness that zoonosis can be acquired from food was highest in Debark and it is more in men than in women and this awareness decreased with increased age of respondents. On the other hand the awareness about foodborne zoonotic disease increased with increased the level of education and income.

**Table 3. Economic impact of foodborne zoonotic diseases on health based on selected households in the community (N = 435).**

| Costs | Cost factors | Gondar | Lalibela | Debark | Total |
|---|---|---|---|---|---|
| Direct Costs | Costs of diagnosis | 6,240.64 | 857.5 | 1,249.5 | 8,347.64 |
| | Costs of drugs | 6,803.16 | 1,964.9 | 3,964.1 | 12,732.16 |
| Indirect costs | CPWTL[1] | 15,000.13 | 3,356.50 | 4,814.25 | 23,170.88 |
| | Costs of informal care takers | 6,750 | 2,145 | 500 | 9,395 |
| Costs of preventive measures | | 57,950 | 2,910 | 6850 | 67,710 |
| Total economic impact | | 92,743.93 | 11,233.90 | 17,377.85 | 121,355.68 |

[1]Costs of productive working time losses.

**Table 4. Descriptive statistics showing awareness of foodborne zoonotic diseases and socio-demographic characteristics of study participants in Amhara region, Ethiopia.**

| Variable | Level | Number of respondents | Aware zoonosis can be acquired from food (%) |
|---|---|---|---|
| Study site | Gondar | 286 | 176 (62) |
| | Lalibela | 71 | 46 (65) |
| | Debark | 78 | 53 (68) |
| Gender | Female | 137 | 80 (58) |
| | Male | 298 | 195 (65) |
| Age in years | <35 | 164 | 100 (61) |
| | 36–50 | 201 | 134 (67) |
| | >50 | 70 | 41 (59) |
| Level of education | Illiterate | 26 | 8 (31) |
| | General education | 54 | 24 (44) |
| | College education | 355 | 243 (68) |
| Household size | One person | 170 | 87 (51) |
| | > one person | 265 | 188 (71) |
| Income category in ETB[1] | < 2500 | 73 | 34 (47) |
| | 2501–5000 | 195 | 111 (57) |
| | 5001–7500 | 124 | 90 (73) |
| | >7500 | 43 | 40 (93) |

[1]Ethiopian Birr

Table 4 summarizes awareness of foodborne zoonosis by different socio-demographic variables.

## 3.5. Socio-demographic factors associated with awareness of foodborne zoonotic diseases

Three variables were statistically significant association with awareness of foodborne zoonotic diseases. Table 5 summarizes univariable associations between awareness of foodborne zoonotic diseases and socio-demographic characteristics of respondents.

Level of education, household size and income were statistically significant association with awareness by the multivariable logistic regression model. The final multivariable model is presented in Table 6.

## 4. Discussion

The objectives of this study were to estimate impact of foodborne zoonotic diseases on health and economy and to assess demographic factors associated with awareness of foodborne zoonotic diseases in Amhara region, Ethiopia. One of the first prerequisites in making decision to implement measures to control foodborne zoonotic diseases is to perceive foodborne zoonotic diseases as a health burden and is an economic problem. Decision makers, however, do not always perceive foodborne zoonotic diseases as costly, and therefore underestimate these diseases as a problem. Although there are a reasonable number of papers on health burden and economic impact of foodborne zoonotic diseases in the literature, to our knowledge, there is no study so far which estimated the economic and health burden of foodborne zoonotic diseases in Ethiopia. Specifically for developing countries this type of knowledge, although valuable, is lacking. Of the 146 studies recently reviewed for major foodborne zoonotic bacterial pathogens, no study estimated the economic and health burden of foodborne zoonotic diseases

**Table 5. Univariable associations between awareness of foodborne zoonotic diseases and socio-demographic characteristics of 435 respondents in Amhara region, Ethiopia.**

| Variable | Level | OR[1] | 95% CI[2] | P-value |
|---|---|---|---|---|
| Study town | Gondar | Ref[3] | | |
| | Lalibela | 1.15 | (0.67–1.98) | 0.613 |
| | Debark | 1.3 | (0.78–2.25) | 0.300 |
| Gender | Female | Ref. | | |
| | Male | 1.35 | (0.89–2.04) | 0.158 |
| Age in years | <35 | Ref. | | |
| | 36–50 | 1.28 | (0.83–1.97) | 0.260 |
| | >50 | 0.90 | (0.51–1.60) | 0.731 |
| Level of education | Illiterate | Ref. | | |
| | High School | 1.8 | (0.67–4.85) | 0.245 |
| | Diploma and above | 4.89 | (2.06–11.56) | 0.001 |
| Household size | One person | Ref. | | |
| | >one person | 2.33 | (1.56–3.48) | 0.001 |
| Income | < 2500 | Ref. | | |
| | 2501–5000 | 1.52 | (0.88–2.60) | 0.131 |
| | 5001–7500 | 3.04 | (1.66–5.57) | 0.001 |
| | >7500 | 15.29 | (4.34–53.93) | 0.001 |

[1]Odds ratio.

[2]Confidence interval.

[3]Reference.

in developing countries [31]. Through estimating economic and health burden of foodborne zoonotic diseases in Ethiopia, the current study shows the need of improving prevention of foodborne zoonotic diseases.

Decision makers are often unaware of the economic impacts of foodborne zoonotic diseases. Creating awareness on the health and economic burden of foodborne zoonotic diseases may be an important step, because people first have to consider foodborne zoonotic diseases as

**Table 6. The final multivariable mixed models describing the associations between awareness of foodborne zoonotic diseases and respondents socio-demographic characteristics on 435 respondents in Amhara region, Ethiopia.**

| Variable | Level | OR[1] | 95% CI[2] | P-value |
|---|---|---|---|---|
| Level of education | Illiterate | Ref[3] | | |
| | High School | 2.30 | (0.83–6.36) | 0.108 |
| | Diploma and above | 5.17 | (1.86–14.39) | 0.002 |
| Household size | One person | Ref. | | |
| | >one person | 2.27 | (1.48–3.49) | 0.001 |
| Income | < 2500 | Ref. | | |
| | 2501–5000 | 0.76 | (0.38–1.530) | 0.448 |
| | 5001–7500 | 1.24 | (0.55–2.78) | 0.600 |
| | >7500 | 5.64 | (1.43–22.32) | 0.014 |

[1]Odds ratio.

[2]Confidence interval.

[3]Reference.

problem before they are motivated to take measure [42]. This may be helpful in their motivation to implement preventive measures that (partly) may have been known already, but that were considered to not be outweighing the problem to be solved. Thus giving more insight to the health and economic burden associated to foodborne zoonotic diseases may motivate households to change their behavior related to prevention of these diseases within their cultural framework for better community acceptance.

Collecting data on factors such as additional cost for better food for sick person and better care under the local circumstances was quite challenging. Disability weights for diarrhea and medication associated to foodborne zoonotic diseases were used from the literature. Since respondents didn't know whether a disease affecting one or more of their household members was zoonotic, the response obtained in the questionnaire was extrapolated towards a whole year. Consequently, the number of foodborne zoonotic diseases was obtained based on extrapolating the data obtained in the questionnaire using confirmatory diagnosis from retrospective data in health records.

The economic impact of foodborne zoonotic diseases in three towns of Amhara region, in average, was 279 ETB per household per year. This means that it was equivalent to 0.5% of the average annual income of respondents. Besides the economic impacts described in this paper, there are additional costs associated to foodborne zoonotic diseases result from costs related to transport costs to travel to hospitals. However, this cost was not included as both patient and the care taker mostly walk to hospitals. This might have underestimated the economic impact of foodborne zoonotic diseases. In absolute terms, the economic impacts of foodborne zoonotic diseases estimated looks small but is significantly related to the average income of Ethiopian respondents. So the study indicated that foodborne zoonotic diseases are causing significant economic impact to households in the study area.

There was large variation in the economic impacts of foodborne zoonotic diseases between households. The largest part of the variation can be explained by differences in costs of preventive measures and in costs of treatment and drug. Although we wanted to base the estimate of economic impacts of foodborne zoonotic diseases as much as possible on actual confirmed data, it was not possible to retrieve all needed data from the field. Therefore we had to make a number of assumptions, for instance on costs of labor. Because of the assumption, there may be discrepancies between the real economic impacts of foodborne zoonotic diseases and our estimates. A relatively large part (52%) of the total economic impacts of foodborne zoonotic diseases was due to costs of preventive measures. However, as there is no any previous study, to our knowledge, it was not possible to compare. The contribution of costs of patients' time to the total economic impacts of foodborne zoonotic diseases is caused by the long time taken for treatment, sick leave, and absenteeism from work because of the disease.

Larger majority of respondents (70.1%) had known the presence of zoonotic diseases but only a smaller majority (63.2%) knew disease can be acquired from food. Both findings were substantially lower than the earlier reported in and around Addis Ababa, Ethiopia [26] although the respondent of that study were elementary and high school students, and Graduates of which 33% were health professionals. Level of education, income and household size were associated with awareness of foodborne zoonotic diseases. This may be related to the fact that more level of education has direct relevance to awareness related health and diseases. Households with more than one person in a house had higher odds of awareness of foodborne zoonotic diseases than households of one person in a house. This one is explained from the fact that if more people live in the household, there is more chance that at least one of them be aware and transmits this awareness to other members of the household.

This study aimed to assess the burden of foodborne zoonotic diseases in Amhara region, Ethiopia in terms of DALYs. The estimates might not have been very accurate as cases were

determined by extrapolation of few confirmed cases based on retrospective data. Moreover, we found that not all cases with diarrhea seek health service. As acknowledged by WHO [43], the high life expectancy allocated and the relatively higher number of cases and death to children result in higher DALYs of foodborne zoonotic diseases in these age groups. Extrapolation to the national scale can provide an indication of the health and economic burden associated to foodborne zoonotic diseases in the country. However, the estimate should be interpreted with caution, given the relative limited number of towns studied and the nature of the data.

## 5. Conclusions

Foodborne zoonotic diseases were associated with high health and economic burden in Amhara Region, Ethiopia. The overall health burden due to foodborne zoonotic diseases (aggregated over the 435 households in Gondar, Lalibela and Debark towns) was estimated to be 89.9 DALYs per 100,000 populations per year. On average, the economic burden was 278.98 ETB per household per year and show large variation between households. Relative to the income of households, these costs are high. The total costs of foodborne zoonotic diseases were mainly due to costs of preventive measures. Although further study is needed at some points, respondents socio-demographic characteristics associated with awareness of foodborne zoonotic diseases were identified, that can be helpful in tailoring control of foodborne zoonotic diseases in Amhara region, Ethiopia. Changing mindset and practical training aiming in controlling foodborne zoonotic diseases is suggested in the health improvement extension service.

## Supporting information

**S1 Questionnaire. Questionnaire on health and economic burden of foodborne zoonotic diseases.**
(DOC)

**S1 Table.**
(CSV)

## Acknowledgments

We gratefully acknowledge respondents who participated in the study. We are also thankful to medical directors who allow us to collect retrospective data from health records.

## Author Contributions

**Conceptualization:** Sefinew Alemu Mekonnen, Adugna Berju, Belete Haile, Haileyesus Dejene, Wassie Molla, Wudu Temesgen Jemberu.

**Data curation:** Agegnehu Gezehagn.

**Formal analysis:** Sefinew Alemu Mekonnen.

**Funding acquisition:** Sefinew Alemu Mekonnen, Adugna Berju, Belete Haile, Haileyesus Dejene, Seleshe Nigatu, Wudu Temesgen Jemberu.

**Investigation:** Sefinew Alemu Mekonnen, Agegnehu Gezehagn, Adugna Berju, Belete Haile, Haileyesus Dejene, Seleshe Nigatu, Wassie Molla, Wudu Temesgen Jemberu.

**Methodology:** Sefinew Alemu Mekonnen, Wudu Temesgen Jemberu.

**Project administration:** Sefinew Alemu Mekonnen, Adugna Berju, Belete Haile, Haileyesus Dejene, Seleshe Nigatu, Wassie Molla, Wudu Temesgen Jemberu.

**Supervision:** Adugna Berju.

**Validation:** Sefinew Alemu Mekonnen, Wudu Temesgen Jemberu.

**Visualization:** Sefinew Alemu Mekonnen, Agegnehu Gezehagn, Adugna Berju, Belete Haile, Haileyesus Dejene, Seleshe Nigatu, Wassie Molla, Wudu Temesgen Jemberu.

**Writing – original draft:** Sefinew Alemu Mekonnen, Agegnehu Gezehagn, Adugna Berju.

**Writing – review & editing:** Sefinew Alemu Mekonnen, Agegnehu Gezehagn, Adugna Berju, Belete Haile, Haileyesus Dejene, Wassie Molla, Wudu Temesgen Jemberu.

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
