## [Decision Letter · Decision Letter 0]

2 Dec 2021

PONE-D-21-22519Health and economic burden of foodborne zoonotic diseases in Amhara region, EthiopiaPLOS ONE

Dear Dr. Mekonnen,

Thank you for submitting your manuscript to PLOS ONE. After careful consideration, we feel that it has merit but does not fully meet PLOS ONE’s publication criteria as it currently stands. Therefore, we invite you to submit a revised version of the manuscript that addresses the points raised during the review process. Please submit your revised manuscript by Jan 16 2022 11:59PM. If you will need more time than this to complete your revisions, please reply to this message or contact the journal office at plosone@plos.org. Please include the following items when submitting your revised manuscript:A rebuttal letter that responds to each point raised by the academic editor and reviewer(s). You should upload this letter as a separate file labeled 'Response to Reviewers'.A marked-up copy of your manuscript that highlights changes made to the original version. You should upload this as a separate file labeled 'Revised Manuscript with Track Changes'.An unmarked version of your revised paper without tracked changes. You should upload this as a separate file labeled 'Manuscript'.

We look forward to receiving your revised manuscript.

Kind regards,

Bijaya Kumar Padhi, PhD, MPH

Academic Editor

PLOS ONE

Journal Requirements:

Additional Editor Comments (if provided):

This manuscript investigates health and economic burden of foodborne zoonotic diseases in Amhara region, Ethiopia. I commend the authors for conceptualizing this study, as the findings certainly add value to the literature. I have following comments:

1) The title of the manuscript should be framed as per STROBE guideline.

2) Psychometrics of the survey instruments should be provided.

3) Sample size calculations including power analysis should be provided

4) Authors said they used multiple logistic regression model. However, there is a lack of clarity on what basis the model was built. Whether they used a-priori hypothesis or any stepwise input of parameters. This information should be provided in data analysis section.

5) Table#5: please correct title of the table as per the data.

Reviewers' comments:

Reviewer's Responses to Questions

**Comments to the Author**

1. Is the manuscript technically sound, and do the data support the conclusions?

Reviewer #1: Yes

Reviewer #2: Yes

2. Has the statistical analysis been performed appropriately and rigorously? 

Reviewer #1: Yes

Reviewer #2: Yes

3. Have the authors made all data underlying the findings in their manuscript fully available?

Reviewer #1: Yes

Reviewer #2: Yes

4. Is the manuscript presented in an intelligible fashion and written in standard English?

Reviewer #1: No

Reviewer #2: Yes

5. Review Comments to the Author

Reviewer #1: Comments on the article reviewed

Material and methods:

• What is meant by projected population?

• Sample size and sampling method not properly documented. How 435 households were selected from 3 towns and method of recruitment of the respondents in the study is not clear.

• The term “haphazardly recruited” is not statistically correct.

• Animal source food consumption habits should be documented in either introduction or results or discussion.

• Grammatical errors are there in the text. It should be written in past tense but at some places, it is written in present or future tense.

• In data management and data analysis, write down the methods/software used e.g. Excel, SPSS etc.

Results

• Did not discuss the socio-demographic profile of the respondents in the starting of the results.

• Have not mentioned the risk factors or food consumption habits here.

Discussion

• Comparison with similar earlier studies is not sufficient.

Overall remarks

• The study findings are good in terms of health and economic burden. But need to rectify the material and methods. Also there is need to pay focus on risk factors and food consumption habits. Also give recommendations based on the findings of the study.

Reviewer #2: line 31- delet is and include 'may be'

line - 97- insert 'literature' after the word no

I had made corrections in the sticky notes and uploaded the pdf paper

6. PLOS authors have the option to publish the peer review history of their article (what does this mean?). If published, this will include your full peer review and any attached files.

Reviewer #1: No

Reviewer #2: **Yes: **AMARENDRA MAHAPATRA

---

## [Author Response · Author response to Decision Letter 0]

11 Dec 2021

We appreciate this thorough review and proper feedback. Indeed, we agree with this reviewer that there are several flaws and we have corrected according to the reviewers’ comments. 

We checked the STROBE guidelines and mentioned the study design in the abstract. 

We highlighted all changes in yellow in the original version of the manuscript and we responded to all specific comments below, indicating line numbers in the revised manuscript where changes were made.

---

## [Editor Report · Decision Letter 1]

16 Dec 2021

Health and economic burden of foodborne zoonotic diseases in Amhara region, Ethiopia

PONE-D-21-22519R1

Dear Dr. Mekonnen,

We’re pleased to inform you that your manuscript has been judged scientifically suitable for publication and will be formally accepted for publication once it meets all outstanding technical requirements.

Kind regards,

Bijaya Kumar Padhi, PhD, MPH

Academic Editor

PLOS ONE
---

## [Editor Report · Acceptance letter]

20 Dec 2021

PONE-D-21-22519R1 

Health and economic burden of foodborne zoonotic diseases in Amhara region, Ethiopia 

Dear Dr. Mekonnen:

I'm pleased to inform you that your manuscript has been deemed suitable for publication in PLOS ONE. Congratulations! Your manuscript is now with our production department. 

Kind regards, 

on behalf of

Dr. Bijaya Kumar Padhi 

Academic Editor

PLOS ONE